# Frictionless Hamiltonian Descent with Discretization and Parallel Optimization

**Heng Zhu**                                                                                     *hez007@ucsd.edu*
*HDSI, University of California San Diego*
**Jun-Kun Wang**                                                                                 *jkw005@ucsd.edu*
*ECE & HDSI, University of California San Diego*

**Reviewed on OpenReview:** *https://openreview.net/forum?id=114IOQ3JWe*

## Abstract

*Frictionless* Hamiltonian Descent is a recently proposed optimization method that leverages a fundamental principle from classical mechanics. The algorithm is based on energy conservation of the Hamiltonian Flow, with resetting the kinetic energy at each iteration, and is shown to be a descent method. However, the idealized frictionless Hamiltonian Descent requires access to the oracle of the Hamiltonian Flow, while exactly implementing the Hamiltonian Flow becomes elusive when the underlying function is not quadratic. Motivated from considerable popularity of Hamiltonian dynamics in sampling, where a geometric numerical integrator is used to simulate the idealized Hamiltonian Monte Carlo, we consider Hamiltonian Descent with two kinds of integrator, which results in some new optimization dynamics. Moreover, we extend the original framework by introducing various forms of kinetic energy. This expansion yields a broad class of optimization algorithms and provides a fresh perspective of algorithm design. We further propose a novel parallelization technique for parallelizing the inherently sequential updates of the proposed optimization algorithms, where gradients at different points are computed simultaneously. The parallelization technique improves the actual running time by 2-3x in practice for multinomial logistic regression across a range of datasets when 4 GPUs is used, compared to approximating the Hamiltonian Flow in the standard sequential fashion by a single GPU.

## 1 Introduction

Hamiltonian mechanics is a fundamental subject in classical mechanics that provides a mechanism to predict the behaviors of many physical systems by unfolding the dynamics over time. Specifically, the Hamiltonian dynamic offers a powerful mathematical framework for describing the dynamic of a particle in phase space that follows the physical laws in Nature. The nice and elegant Hamiltonian perspective exposes important features of the systems such as energy conservation, and it also allows for a deeper study because of its rich mathematical structure. In machine learning and computational statistics, the applications of the Hamiltonian dynamics principle lead to the classical Hamiltonian Monte Carlo algorithm (HMC) (Duane et al., 1987; Neal, 2011; Durmus et al., 2017; Wang & Wibisono, 2023) and its numerous variants (Kook et al., 2022; Lee et al., 2021; Noble et al., 2024; Mangoubi & Vishnoi, 2018; Ver Steeg & Galstyan, 2021; Hoffman & Gelman, 2014; Chaari et al., 2016), which are for the task of sampling and have been found useful in many other fields such as systems biology (Kook et al., 2022).

Recently the principle of Hamiltonian mechanics has inspired designing optimization algorithms for solving an optimization problem, i.e., solving $\min_{\mathbf{x} \in \mathbb{R}^d} f(\mathbf{x})$ (see e.g., Wang (2024); Diakonikolas & Jordan (2021); Teel et al. (2019)). Specifically, Wang (2024) proposes a new optimization algorithm called *frictionless* Hamiltonian Descent. They define a Hamiltonian $H(\mathbf{x}, \mathbf{v})$ as $H(\mathbf{x}, \mathbf{v}) := f(\mathbf{x}) + \frac{1}{2}\|\mathbf{v}\|_2^2$, where $\mathbf{x}$ is the position of the update, and $\mathbf{v}$ is its velocity. The function $f(\cdot)$ in their definition of the Hamiltonian is the underlying function being optimized. From the perspective of Hamiltonian mechanics, the function value

---

**Algorithm 1** *Idealized Frictionless* HAMILTONIAN DESCENT (IDEALIZED-HD) (Wang, 2024)

---

1: **Input:** an initial point $\mathbf{x}_1 \in \mathbb{R}^d$, number of iterations $K$, and a scheme of integration time $(\eta_k)_{k=1}^K$.
2: **for** $k = 1$ to $K$ **do**
3:    // *Execute Hamiltonian Flow* (1) *for integration time* $\eta_k$ *from the initial position* $\mathbf{x}_k \in \mathbb{R}^d$ *with the initial velocity* $0 \in \mathbb{R}^d$ *to obtain* $\mathbf{x}_{k+1}$, *i.e., the position of the particle at time* $t = \eta_k$.
4:    $(\mathbf{x}_{k+1}, \mathbf{v}_{k+1}) = \mathsf{HF}_{\eta_k}(\mathbf{x}_k, 0)$.
5: **end for**
6: **Output:** $\mathbf{x}_{K+1}$.

---

$f(\mathbf{x})$ is the potential energy of one particle with its position $\mathbf{x} \in \mathbb{R}^d$, while the term $\frac{1}{2}\|\mathbf{v}\|_2^2$ in the Hamiltonian is the kinetic energy. In other words, the total energy $H(\mathbf{x}, \mathbf{v})$ of a particle consists of its potential energy and its kinetic energy. With a specified Hamiltonian, the corresponding Hamiltonian Flow is described by a system of differential equations:

$$\textbf{(Hamiltonian Flow)} \quad \frac{d\mathbf{x}}{dt} = \frac{\partial H}{\partial \mathbf{v}} \text{ and } \frac{d\mathbf{v}}{dt} = -\frac{\partial H}{\partial \mathbf{x}}. \tag{1}$$

Wang (2024) observes that by energy conservation of the Hamiltonian flow, i.e., $H(\mathbf{x}_t, \mathbf{v}_t) = H(\mathbf{x}_0, \mathbf{v}_0)$ at any time $t$, which is a well-known principle in classical mechanics (Arnold, 1989; Greiner, 2003), it follows

$$f(\mathbf{x}_t) + \frac{1}{2}\|\mathbf{v}_t\|_2^2 = H(\mathbf{x}_t, \mathbf{v}_t) = H(\mathbf{x}_0, \mathbf{v}_0) = f(x_0), \tag{2}$$

when the initial velocity of the flow is zero, i.e., $\mathbf{v}_0 = 0 \in \mathbb{R}^d$. Hence, the mechanism can be used to design a *descent* method, which is Idealized Frictionless Hamiltonian Descent (Idealized-HD) (Algorithm 1). Idealized HD is a direct counterpart of Idealized-HMC in optimization, as the only difference between them is that the initial velocity for executing a Hamiltonian Flow is set to be a random vector from the normal distribution in HMCs. Wang (2024) analyzes Idealized-HD for the class of strongly convex quadratic problems, where exactly simulating the Hamiltonian Flow is feasible, as the Hamiltonian system (1) has a closed-form solution for this specific class of functions. It also proposes Idealized Frictionless Coordinate Hamiltonian Descent (CHD), which updates a coordinate in each iteration, and shows that classical algorithms such as Gauss-Seidel methods, Successive Over-relaxation, Jacobi method, and weighted Jacobi method for solving linear systems of equations are all instances of CHD. Precisely, they are CHD with different schemes of the integration time for executing the Hamiltonian Flow. The optimization mechanism of running a *frictionless* Hamiltonian Flow with resets by Wang (2024) is novel in the sense that it challenges the conventional wisdom in optimization, which believes that incorporating a friction term into designing the optimization dynamics can facilitate the optimization progress. Specifically, a common approach in the optimization literature is to include a *friction* term $\gamma\mathbf{v}$ into the update, e.g., $\frac{d\mathbf{v}}{dt} = -\frac{\partial H}{\partial \mathbf{x}} - \gamma\mathbf{v}$, where $\gamma > 0$ serves as a damping factor to dissipate energy during the flow. This dissipation underpins various momentum-based methods, e.g., Nesterov (1983); Polyak (1964); O'Donoghue & Maddison (2019); Wang & Abernethy (2018); Su et al. (2016); Maddison et al. (2018). However, Wang (2024) demonstrates that even without the friction term, the Idealized Hamiltonian Descent (Idealized-HD) achieves an accelerated convergence rate as those of friction-based accelerated methods (Nesterov, 1983; Polyak, 1964; Wang & Abernethy, 2018; Su et al., 2016) for strongly convex quadratic functions. We further note that recently, Fu & Wibisono (2025) studied frictionless Hamiltonian Descent with randomized integration time and demonstrated that their proposed scheme enables Hamiltonian Descent to achieve the provable accelerated rate in general strongly convex optimization.

While the results presented in Wang (2024) are neat, the Idealized-HD requires the exact Hamiltonian Flow, and hence whether it is realizable for minimizing general non-quadratic functions is not clear, since the differential equation system (1) does not have a closed-form solution in general. Motivated from the area of sampling (Mangoubi & Smith, 2017; Chen & Gatmiry, 2023; Chen et al., 2020; Durmus et al., 2017; Mangoubi & Vishnoi, 2018; Monmarché, 2022; Camrud et al., 2023; Bou-Rabee & Sanz-Serna, 2018), we consider two geometric integrators to numerically approximate the Hamiltonian Flow; one is the Störmer-Verlet integrator (Hairer et al., 2003; Ernst Hairer, 2006) (a.k.a. the Leapfrog integrator) and the other is

a recently proposed stratified Monte Carlo integrator (Bou-Rabee & Marsden, 2022). When applying these integrators to approximate the Hamiltonian Flow in Hamiltonian Descent, the resulting updates exhibit novel momentum-like dynamics, where the velocity $\mathbf{v}$ of the update $\mathbf{x}$ is an accumulation of gradients along the discretized Hamiltonian Flow, with a reset after each flow. Interestingly, the use of the Leapfrog leads to an optimization dynamic that computes the gradients at the points along the flow, while the stratified Monte Carlo integrator generates a dynamic where the gradient is computed at a point located a certain distance from the update, determined by a random step in the direction of the current velocity.

We also extend the framework of optimization via Hamiltonian mechanics in Wang (2024) to a more general case. More precisely, we consider a more general form of the Hamiltonian in our work:

$$H(\mathbf{x}, \mathbf{v}) := f(\mathbf{x}) + \phi(\mathbf{v}), \tag{3}$$

where the kinetic energy $\phi(\cdot)$ (to be specified) satisfies $\phi(\mathbf{v}) \geq 0, \forall \mathbf{v} \in \mathbb{R}^d$ and $\phi(\mathbf{v}) = 0$ if and only if the velocity $\mathbf{v} = 0$. We found that under different forms of the kinetic energy $\phi(\mathbf{v})$, the resulting dynamics of Hamiltonian Descent with the Leapfrog integrator lead to different known and new updates, including Gradient Descent (GD), Normalized Gradient Descent (Normalized GD), Sign Gradient Descent (Sign GD), Coordinate Descent (CD) as well as their counterparts that incorporate certain novel momentum-like dynamics. Specifically, while the use of a single integrator step to simulate the Hamiltonian Flow at each iteration of Hamiltonian Descent produces these known algorithms, executing multiple steps of the integrator induces some new dynamics that update the current position of the iterate via the gradients collected since the last reset of the velocity.

We further propose a method to speed up simulating the Hamiltonian Flow via simultaneously computing the gradients at some points along the discretized flow. Specifically, given access to a computational environment with multiple GPUs, we aim to accelerate the process of numerically simulating a Hamiltonian Flow by having each GPU compute a gradient at different steps of the flow in parallel and in an iterative fashion. Although the Hamiltonian Flow is sequential by nature, i.e., a gradient at the current position is used to update the iterate before computing the gradient at the next step, we found that via the underlying mechanism of Picard Iteration (Teschl, 2012; Khalil, 2002; Andrade et al., 2023), computing gradients at different steps in parallel becomes feasible, which in turn leads to a faster optimization process in practice. We conduct experiments on the proposed Hamiltonian Descent via Parallel Picard Iteration for solving multinomial logistic regression across multiple datasets. We found that the improvement of the speedup in running time is strikingly consistent across all the datasets. For example, compared to the case of using a single GPU, the parallel optimization with 4 GPUs reduces the actual running time of simulating a Hamiltonian Flow by 2.13x on a8a dataset and by 3.23x on MNIST dataset.

## 2 Frictionless Hamiltonian Descent with Geometric Numerical Integrators

We begin by observing that the idea of optimization via Hamiltonian mechanics in Wang (2024) can be more applicable when considering the Hamiltonian in its general form (3). Indeed, the energy conservation along the Hamiltonian Flow (1) still holds.

**Lemma 1.** *(See e.g., Arnold (1989); Greiner (2003)) The time derivative of a time-independent Hamiltonian is $0$ along the Hamiltonian Flow (1), i.e., $\frac{dH}{dt} = 0$.*

*Proof.* $\frac{dH}{dt} = \left\langle \frac{\partial H}{\partial \mathbf{x}}, \frac{\partial \mathbf{x}}{\partial t} \right\rangle + \left\langle \frac{\partial H}{\partial \mathbf{v}}, \frac{\partial \mathbf{v}}{\partial t} \right\rangle = \left\langle \frac{\partial H}{\partial \mathbf{x}}, \frac{\partial H}{\partial \mathbf{v}} \right\rangle + \left\langle \frac{\partial H}{\partial \mathbf{v}}, -\frac{\partial H}{\partial \mathbf{x}} \right\rangle = 0.$ □

Using Lemma 1 and that the kinetic energy is zero when the velocity is zero, i.e., $\phi(0) = 0$, it becomes evident that Idealized-HD (Algorithm 1) with the general notion of the Hamiltonian (3) satisfies:

$$\begin{aligned} f(\mathbf{x}_{k+1}) + \phi(\mathbf{v}_{k+1}) =& H(\mathbf{x}_{k+1}, \mathbf{v}_{k+1}) \\ =& H(\mathbf{x}_k, 0) = f(\mathbf{x}_k), \quad \forall k, \end{aligned} \tag{4}$$

---

**Algorithm 2** FRICTIONLESS HAMILTONIAN DESCENT WITH A GEOMETRIC NUMERICAL INTEGRATOR.

---

1: **Input:** an initial point $\mathbf{x}_0 \in \mathbb{R}^d$, number of iterations $K$, a scheme of integration time $(\eta_k)_{k=1}^K$, and the step size used in a geometric numerical integrator $\theta$.
2: Specify the Hamiltonian $H(\mathbf{x}, \mathbf{v})$.
3: **for** $k = 1$ to $K$ **do**
4:     Obtain $(\mathbf{x}_{k+1}, \mathbf{v}_{k+1}) = \textsc{Integrator}_{S_k, \theta}(\mathbf{x}_k, 0)$, where $S_k := \lfloor \frac{\eta_k^{(K)}}{\theta} \rfloor$.
    *// Numerically simulate the Hamiltonian Flow, where* INTEGRATOR *can be, for example,* LeapFrog *(Algorithm 3) or* Stratified *(Algorithm 4).*
5: **end for**
6: **Output:** $\mathbf{x}_{k+1}$.

---

where the kinetic energy is nonnegative, i.e., $\phi(\mathbf{v}_{k+1}) \geq 0$. Equation (4) reduces to (2) when $\phi(\cdot) = \frac{1}{2}\|\cdot\|_2^2$. On the other hand, it also encapsulates the cases when the kinetic energy $\phi(\cdot)$ is in a different form. For example, in the following, we will also consider $\phi(\mathbf{v}) = \|\mathbf{v}\|_2, \|\mathbf{v}\|_1$, and $\|\mathbf{v}\|_\infty$.

As described in the introduction, the differential equation system (1) does not have a closed-form solution in general, and one may need to use a geometric numerical integrator to simulate Hamiltonian flow. We hence propose Algorithm 2, which utilizes an integrator to simulate the continuous-time dynamic in discrete time. We note that approximating the flow via a geometric numerical integrator is standard in sampling (Mangoubi & Smith, 2017; Chen & Gatmiry, 2023; Chen et al., 2020; Durmus et al., 2017; Mangoubi & Vishnoi, 2018; Monmarché, 2022; Camrud et al., 2023; Bou-Rabee & Sanz-Serna, 2018), and perhaps one of the most popular integrators is the Leapfrog integrator (Hairer et al., 2003; Ernst Hairer, 2006). We thus consider Hamiltonian Descent (HD) (Algorithm 2) with the Leapfrog integrator first, i.e., let INTEGRATOR on Line 4 in Algorithm 2 be LeapFrog (Algorithm 3). We note that since Leapfrog integrator is a symplectic integrator, thus not exactly conserving the Hamiltonian or total energy. However, it is still widely used in sampling area due to its bounded energy error. The energy oscillates near the true Hamiltonian at the order $O(\theta^2)$, where $\theta$ is the time step size (Neal, 2011; Ernst Hairer, 2006). Unlike other non-symplectic integrators which brings systematic energy drift, the energy error of Leapfrog integrator grows slowly with time, making it a good choice to approximate the continuous Hamiltonian dynamic.

In this paper we use $\bar{\mathbf{x}}$ and $\bar{\mathbf{v}}$ to denote the variables inside one iteration. It turns out that when a single Leapfrog step (i.e., $S = 1$) is used to simulate the Hamiltonian Flow at each iteration of HD, the resulting dynamics recover Gradient Descent, Normalized Gradient Descent, Sign Gradient Descent, and Coordinate Descent, under different forms of the kinetic energy. We summarize the results in Lemma 2.

---

**Algorithm 3** THE LEAPFROG INTEGRATOR $\mathsf{LeapFrog}_{S,\theta}(\bar{\mathbf{x}}_0, \bar{\mathbf{v}}_0)$.

---

1: **Input:** an initial point $(\bar{\mathbf{x}}_0, \bar{\mathbf{v}}_0)$, # of Leapfrog steps $S$, and the step size $\theta > 0$.
2: **for** $s = 0$ to $S - 1$ **do**
3:     $\bar{\mathbf{v}}_{s+\frac{1}{2}} = \bar{\mathbf{v}}_s - \frac{\theta}{2}\nabla_\mathbf{x} H(\bar{\mathbf{x}}_s, \bar{\mathbf{v}}_s)$.
4:     $\bar{\mathbf{x}}_{s+1} = \bar{\mathbf{x}}_s + \theta \, \partial_\mathbf{v} H\left(\bar{\mathbf{x}}_s, \bar{\mathbf{v}}_{s+\frac{1}{2}}\right)$.
5:     $\bar{\mathbf{v}}_{s+1} = \bar{\mathbf{v}}_{s+\frac{1}{2}} - \frac{\theta}{2}\nabla_\mathbf{x} H(\bar{\mathbf{x}}_{s+1}, \bar{\mathbf{v}}_{s+\frac{1}{2}})$.
6: **end for**
7: **Output:** $\mathsf{LeapFrog}_{S,\theta}(\bar{\mathbf{x}}_0, \bar{\mathbf{v}}_0) := (\bar{\mathbf{x}}_S, \bar{\mathbf{v}}_S)$.

---

**Algorithm 4** THE STRATIFIED INTEGRATOR $\mathsf{Stratified}_{S,\theta}(\bar{\mathbf{x}}_0, \bar{\mathbf{v}}_0)$. (same input as Alg. 3)

---

1: **for** $s = 0$ to $S - 1$ **do**
2:     $\bar{\mathbf{x}}_{s+\frac{1}{2}} = \bar{\mathbf{x}}_s + \zeta_s \, \partial_\mathbf{v} H(\bar{\mathbf{x}}_s, \bar{\mathbf{v}}_s)$, where $\zeta_s \sim$ **Uniform**$(0, \theta)$.
3:     $\bar{\mathbf{v}}_{s+\frac{1}{2}} = \bar{\mathbf{v}}_s - \frac{\theta}{2}\nabla_\mathbf{x} H(\bar{\mathbf{x}}_{s+\frac{1}{2}}, \bar{\mathbf{v}}_s)$.
4:     $\bar{\mathbf{x}}_{s+1} = \bar{\mathbf{x}}_s + \theta \, \partial_\mathbf{v} H\left(\bar{\mathbf{x}}_s, \bar{\mathbf{v}}_{s+\frac{1}{2}}\right)$.
5:     $\bar{\mathbf{v}}_{s+1} = \bar{\mathbf{v}}_{s+\frac{1}{2}} - \frac{\theta}{2}\nabla_\mathbf{x} H(\bar{\mathbf{x}}_{s+\frac{1}{2}}, \bar{\mathbf{v}}_s)$.
6: **end for**
7: **Output:** $\mathsf{Stratified}_{S,\theta}(\bar{\mathbf{x}}_0, \bar{\mathbf{v}}_0) := (\bar{\mathbf{x}}_S, \bar{\mathbf{v}}_S)$.

---

**Lemma 2.** *Consider Frictionless Hamiltonian Descent with the Leapfrog integrator (HD-LF), i.e., Algorithm 2 with* INTEGRATOR = LeapFrog. *Set a single Leapfrog step $S_k = 1$ at each iteration $k$. Then, we have*

1. *Setting $\phi(\mathbf{v}) = \frac{1}{2}\|\mathbf{v}\|_2^2$ recovers GRADIENT DESCENT (GD):*

$$\mathbf{x}_{k+1} = \mathbf{x}_k - \frac{\theta^2}{2}\nabla f(\mathbf{x}_k).$$

2. *Setting $\phi(\mathbf{v}) = \|\mathbf{v}\|_2$ recovers NORMALIZED GD:*

$$\mathbf{x}_{k+1} = \mathbf{x}_k - \theta\frac{\nabla f(\mathbf{x}_k)}{\|\nabla f(\mathbf{x}_k)\|}.$$

3. *Setting $\phi(\mathbf{v}) = \|\mathbf{v}\|_1$ recovers SIGN GD:*

$$\mathbf{x}_{k+1} = \mathbf{x}_k - \theta\,\mathrm{sign}(\nabla f(\mathbf{x}_k)).$$

4. *Setting $\phi(\mathbf{v}) = \|\mathbf{v}\|_\infty$ recovers COORDINATE DESCENT (CD):*

$$\mathbf{x}_{k+1} = \mathbf{x}_k - \theta\,\mathrm{sign}\left(\nabla f(\mathbf{x}_k)[i_{\max}]\right)\mathbf{e}_{i_{\max}},$$

*where $i_{\max} \in \arg\max_{i\in[d]}|\nabla f(\mathbf{x}_k)[i]|$.*

*Proof.* We provide the proof of the first item here and defer the proof of the rest of the items to Appendix A.

When $\phi(\mathbf{v}) = \frac{1}{2}\|\mathbf{v}\|_2^2$, we have $\nabla_{\mathbf{x}}H(\mathbf{x},\mathbf{v}) = \nabla f(\mathbf{x})$ and $\nabla_{\mathbf{v}}H(\mathbf{x},\mathbf{v}) = \mathbf{v}$. From Algorithm 3, with the initial condition $\bar{\mathbf{x}}_0 \leftarrow \mathbf{x}_k$ and $\bar{\mathbf{v}}_0 \leftarrow 0$, one can see that the dynamic with a single Leapfrog step is:

$$\bar{\mathbf{v}}_{\frac{1}{2}} = \bar{\mathbf{v}}_0 - \frac{\theta}{2}\nabla f(\bar{\mathbf{x}}_0) = -\frac{\theta}{2}\nabla f(\mathbf{x}_k)$$

$$\bar{\mathbf{x}}_1 = \bar{\mathbf{x}}_0 + \theta\,\bar{\mathbf{v}}_{\frac{1}{2}} = \mathbf{x}_k - \frac{\theta^2}{2}\nabla f(\mathbf{x}_k). \tag{5}$$

With only a single step, Leapfrog outputs $\bar{\mathbf{x}}_1$, which is the position of the update $\mathbf{x}_{k+1}$ at the next iteration $k+1$. We hence obtain the update of GD, i.e., $\mathbf{x}_{k+1} = \mathbf{x}_k - \frac{\theta^2}{2}\nabla f(\mathbf{x}_k)$.

$\square$

On the other hand, when more than a single iteration is executed in Leapfrog, the dynamics differ from the known ones. For example, under the form of the kinetic energy $\phi(\mathbf{v}) = \frac{1}{2}\|\mathbf{v}\|_2^2$, the resulting dynamic with $S$ Leapfrog steps is

$$\bar{\mathbf{x}}_S = \bar{\mathbf{x}}_0 - \frac{S\theta^2}{2}\nabla f(\bar{\mathbf{x}}_0) - \theta^2\sum_{i=1}^{S-1}(S-i)\nabla f(\bar{\mathbf{x}}_i). \tag{6}$$

In addition, when the form of the kinetic energy $\phi(\mathbf{v}) = \|\mathbf{v}\|_2$, the resulting dynamic with $S$ Leapfrog steps is

$$\bar{\mathbf{x}}_S = \bar{\mathbf{x}}_0 - \theta\frac{\nabla f(\bar{\mathbf{x}}_0)}{\|\nabla f(\bar{\mathbf{x}}_0)\|} - \theta\sum_{i=1}^{S-1}\frac{\frac{1}{2}\nabla f(\bar{\mathbf{x}}_0) + \sum_{j=1}^{i}\nabla f(\bar{\mathbf{x}}_j)}{\|\frac{1}{2}\nabla f(\bar{\mathbf{x}}_0) + \sum_{j=1}^{i}\nabla f(\bar{\mathbf{x}}_j)\|}. \tag{7}$$

Furthermore, when the form of the kinetic energy $\phi(\mathbf{v}) = \|\mathbf{v}\|_1$, the resulting dynamic with $S$ Leapfrog steps is

$$\bar{\mathbf{x}}_S = \bar{\mathbf{x}}_0 - \theta\mathrm{sign}(\nabla f(\bar{\mathbf{x}}_0))$$

$$- \theta\sum_{i=1}^{S-1}\mathrm{sign}\left(\frac{1}{2}\nabla f(\bar{\mathbf{x}}_0) + \sum_{j=1}^{i}\nabla f(\bar{\mathbf{x}}_j)\right). \tag{8}$$

Under the form of the kinetic energy $\phi(\mathbf{v}) = \|\mathbf{v}\|_\infty$, the resulting dynamic with $S$ Leapfrog steps is

$$\bar{\mathbf{x}}_S = \bar{\mathbf{x}}_0 - \theta \mathrm{sign}\left(\nabla f(\bar{\mathbf{x}}_0)[i^0_{\max}]\right) \mathbf{e}_{i^0_{\max}}$$
$$- \theta \sum_{i=1}^{S-1} \mathrm{sign}\left[\left(\frac{1}{2}\nabla f(\bar{\mathbf{x}}_0) + \sum_{j=1}^{i} \nabla f(\bar{\mathbf{x}}_j)\right)[i^{\alpha_i}_{\max}]\right] \mathbf{e}^{\alpha_i}_{i_{\max}}. \tag{9}$$

where $i^0_{\max}$ is the index corresponding to the largest absolute value of $\nabla f(\bar{\mathbf{x}}_0)$, and $i^{\alpha_i}_{\max}$ is the index corresponding to the largest absolute value of vector $\left(\frac{1}{2}\nabla f(\bar{\mathbf{x}}_0) + \sum_{j=1}^{i}\nabla f(\bar{\mathbf{x}}_j)\right)$.

In optimization literature, Heavy Ball (Polyak, 1964) is one of the popular momentum methods. After unfolding its dynamic over $S$ steps, one finds

$$\bar{\mathbf{y}}_S = \bar{\mathbf{y}}_0 - \eta \sum_{i=0}^{S-1}\sum_{j=0}^{i} \beta^{i-j}\nabla f(\bar{\mathbf{y}}_j), \tag{10}$$

where $\eta$ is the step size and $\beta$ is the momentum parameter. By comparing the dynamics, (6) and (10), we see that simulating a single Hamiltonian Flow in HD via Leapfrog yields a certain momentum-like dynamic with the momentum parameter $\beta = 1$ (i.e., without reducing the weights of the accumulated gradients). On the other hand, at the end of the flow in HD, the velocity $\mathbf{v}$ and hence the kinetic energy are reset to 0. The reset of the kinetic energy in HD is necessary to make the dynamical system to be dissipative in order to converge. Without the reset, by energy conservation of the Hamiltonian Flow, the update might continue moving past a desired point and hence fail to converge. This mechanism of HD differs from existing *friction-based* momentum methods (Maddison et al., 2018; O'Donoghue & Maddison, 2019; De Luca & Silverstein, 2022; França et al., 2020; Mai & Johansson, 2020; Wibisono et al., 2016; Wilson et al., 2021), which typically involve a movement with friction to dissipate kinetic energy, e.g., via specifying the value of the momentum parameter $\beta$ to be $\beta < 1$.

We further consider Hamiltonian Descent (HD) (Algorithm 2) with the recently proposed Stratified Monte Carlo integrator (Bou-Rabee & Marsden, 2022) (HD-ST), i.e., instantiate INTEGRATOR on Line 4 in Algorithm 2 as Stratified (Algorithm 4). The Stratified Monte Carlo integrator was shown to outperform the Leapfrog integrator for HMC in sampling (Bou-Rabee & Marsden, 2022). When the kinetic energy is in the form $\phi(\mathbf{v}) = \frac{1}{2}\|\mathbf{v}\|_2^2$, the dynamic of HD-ST becomes:

$$\bar{\mathbf{x}}_{s+\frac{1}{2}} = \bar{\mathbf{x}}_s + \zeta_s \bar{\mathbf{v}}_s, \quad \bar{\mathbf{v}}_{s+\frac{1}{2}} = \bar{\mathbf{v}}_s - \frac{\theta}{2}\nabla f(\bar{\mathbf{x}}_{s+\frac{1}{2}}),$$
$$\bar{\mathbf{x}}_{s+1} = \bar{\mathbf{x}}_s + \theta \bar{\mathbf{v}}_{s+\frac{1}{2}} \quad \bar{\mathbf{v}}_{s+1} = \bar{\mathbf{v}}_{s+\frac{1}{2}} - \frac{\theta}{2}\nabla f(\bar{\mathbf{x}}_{s+\frac{1}{2}}). \tag{11}$$

Therefore, different from HD-LF, the gradient at each integrator step is computed at a different point than $\bar{\mathbf{x}}_s$ in HD-ST. More precisely, the point where the gradient is computed is determined by a random step $\zeta_s$ in the direction of the current velocity $\bar{\mathbf{v}}_s$ from $\bar{\mathbf{x}}_s$. Interestingly, when only a single integrator step is executed in each iteration $k$, the resulting dynamics recover the known ones as Lemma 2, albeit with a random step size $\zeta_s$.

## 3  Speeding Up Hamiltonian Descent via Parallel Optimization

One challenge in simulating a continuous-time Hamiltonian Flow via discretization is that while reducing the step size $\theta$ brings the approximated flow closer to the exact Hamiltonian flow, it comes at the expense of requiring more steps in each iteration, as the number of steps is $S_k = \lfloor \eta_k/\theta \rfloor$ at $k$ given a fixed integration time $\eta_k$, thereby might increase the time needed to complete an iteration of HD. To tackle this issue, we aim to answer the following question: can we run the steps in an approximated Hamiltonian Flow in parallel, given access to multiple GPUs that allow computing gradients at different points simultaneously? The answer is affirmative, and our method to achieve the goal is based on a parallel implementation of Picard Iteration.

Picard Iteration is a classical algorithm that oftentimes serves as a technique to prove the existence of the unique solution of a differential equation, $\frac{d\mathbf{x}}{dt} = Q(\mathbf{x}, t)$, where the underlying function $Q(\cdot, \cdot)$ is continuous and $L_0$-Lipschitz, i.e., $|Q(\mathbf{x}, t) - Q(\mathbf{y}, t)| \leq L_0 \|\mathbf{x} - \mathbf{y}\|$ (see e.g., Khalil (2002); Teschl (2012)). Specifically, at each iteration $m$ of Picard Iteration, the update is

$$\textbf{(Picard Iteration)} \qquad \mathbf{x}_t^{(m+1)} = \mathbf{x}_0 + \int_0^t Q(\mathbf{x}_r^{(m)}, r)dr. \tag{12}$$

However, evaluating the integral might not be possible in general, and one may need to resort to numerical computations. Specifically, a way to numerically approximate the update (12) is by computing $\mathbf{x}_s^{(m+1)} = \mathbf{x}_0 + \theta \sum_{j=0}^{s-1} Q(\mathbf{x}_j^{(m)}, j\theta)$, where $\theta > 0$ is the step size. The update $\mathbf{x}_s^{(m+1)}$ can be viewed as a discrete-time approximation to $\mathbf{x}_t^{(m+1)}$ of (12) in continuous time, with $t = s\theta$.

We now consider Picard Iteration for simulating Hamiltonian flow (1) and present here the case when the kinetic energy is $\phi(\cdot) = \frac{1}{2}\|\cdot\|_2^2$. From (1) and (12), the update can be shown to be:

$$\mathbf{v}_t^{(m+1)} = \mathbf{v}_0 - \int_0^t \nabla f(\mathbf{x}_r^{(m)})dr$$

$$\mathbf{x}_t^{(m+1)} = \mathbf{x}_0 + t\mathbf{v}_0 - \int_0^t (t-r)\nabla f(\mathbf{x}_r^{(m)})dr. \tag{13}$$

Then, by discretizing the update scheme (13), we obtain

$$\bar{\mathbf{v}}_s^{(m+1)} = \bar{\mathbf{v}}_0 - \theta \sum_{i=0}^{s-1} \nabla f(\bar{\mathbf{x}}_i^{(m)}), \quad \forall s \in [S],$$

$$\bar{\mathbf{x}}_s^{(m+1)} = \bar{\mathbf{x}}_0 + (s\theta)\bar{\mathbf{v}}_0 - \theta \sum_{i=0}^{s-1} (s-i)\theta \nabla f(\bar{\mathbf{x}}_i^{(m)}), \tag{14}$$

where $\theta$ is the step size, $\eta$ is the integration time of the Hamiltonian flow, and $S = \lfloor \eta/\theta \rceil$ is number of steps to simulate the flow. We further note that the initial velocity in HD is zero (i.e., $\bar{\mathbf{v}}_0 = 0$). The update scheme in (14) implies that one can execute the steps in the flow in parallel. More precisely, if one would like to simulate the flow for $S$ steps, then all $\bar{\mathbf{x}}_s^{(0)}$'s can be initialized randomly at the beginning (i.e., $m = 0$). Furthermore, at each iteration $m$, gradients at different steps $s$ in the flow, i.e., $\nabla f\left(\bar{\mathbf{x}}_s^{(m)}\right)$, can be computed in parallel, and obtaining $\bar{\mathbf{x}}_s^{(m+1)}$ from $\bar{\mathbf{x}}_s^{(m)}$ can also be done in a parallel fashion, which involves the gradients at those $\bar{\mathbf{x}}_i^{(m)}$'s, where $i < s$. We refer the reader to Figure 1 for the illustration of this parallel update scheme. It is known that when $m = S$, then $\bar{\mathbf{x}}_S^{(S)}$ is guaranteed to be exactly that of sequentially computing the update for $S$ steps, i.e., $\bar{\mathbf{x}}_S^{(S)} = \bar{\mathbf{x}}_S$. We formalize this argument with the following lemma. Empirically, we found that the number of iterations $m$ does not need to be as large as $S$ to achieve competitive performance as that of $S$ steps.

**Lemma 3.** *When the number of Picard iteration is $m = S$, then the update of Picard iteration $\bar{\mathbf{x}}_S^{(S)}$ is exactly the output of sequentially computing the discretized Hamiltonian flow after $S$ steps, i.e., $\bar{\mathbf{x}}_S^{(S)} = \bar{\mathbf{x}}_S$.*

*Proof.* We prove this argument by induction. The strategy is to show that at step $s$, we have $\bar{\mathbf{x}}_s^{(j)} = \bar{\mathbf{x}}_s$ for all $j \geq s$.

For the base case $s = 1$, the condition is true since for all $j \geq 1$, we have

$$\bar{\mathbf{x}}_1^{(j)} = \bar{\mathbf{x}}_0 - \theta^2 \nabla f(\bar{\mathbf{x}}_0^{(j)}) = \bar{\mathbf{x}}_0 - \theta^2 \nabla f(\bar{\mathbf{x}}_0) = \bar{\mathbf{x}}_1.$$

Now assume that at a particular step $s > 1$, we have $\bar{\mathbf{x}}_s^{(j)} = \bar{\mathbf{x}}_s$ for all $j \geq s$. Then, at next step $s + 1$, for all $j' \geq s + 1$, we have

$$\bar{\mathbf{x}}_{s+1}^{(j')} = \bar{\mathbf{x}}_0 - \theta^2 \sum_{i=0}^{s} (s+1-i)\nabla f(\bar{\mathbf{x}}_i^{(j'-1)}) = \bar{\mathbf{x}}_0 - \theta^2 \sum_{i=0}^{s} (s+1-i)\nabla f(\bar{\mathbf{x}}_i) = \bar{\mathbf{x}}_s,$$

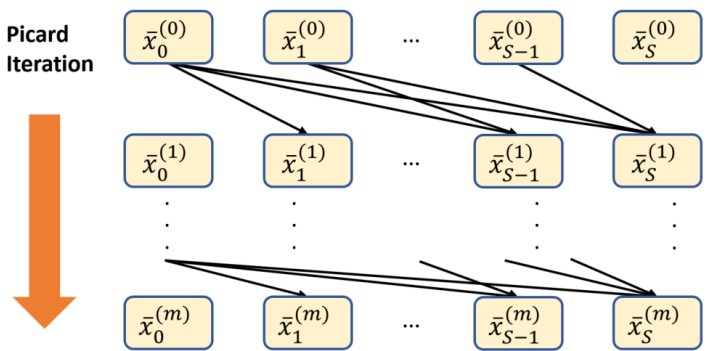

Figure 1: Hamiltonian Flow via Parallel Picard Iteration. The update $\bar{\mathbf{x}}_s^{(m)}$ receives gradients from the preceding ones $\bar{\mathbf{x}}_i^{(m-1)}$, $\forall i < s$, which can be done in parallel.

since $j' \geq s + 1 > s$. Therefore, the condition holds at step $s + 1$, which completes the induction. That is, for all $j \geq S$, we have $\bar{\mathbf{x}}_S^{(j)} = \bar{\mathbf{x}}_S$, which also implies that $\bar{\mathbf{x}}_S^{(S)} = \bar{\mathbf{x}}_S$. □

---

**Algorithm 5** FRICTIONLESS HAMILTONIAN DESCENT VIA PARALLEL PICARD ITERATION

---

1: **Input:** an initial point $\mathbf{x}_1 \in \mathbb{R}^d$, number of iterations $K$, a scheme of integration time $(\eta_k)_{k=1}^K$, a scheme of step sizes $(\theta_k)_{k=1}^K$, a window size $W$, and a threshold parameter $\tau$.
2: **for** $k = 1$ to $K$ **do**
3:     *// Numerically simulate the Hamiltonian Flow $(\mathbf{x}_{k+1}, \cdot) = \mathsf{HF}_{\eta_k}(\mathbf{x}_k, 0)$.*
4:     Set $s = 0, m = 0$.
5:     Init $\bar{\mathbf{x}}_{s+w-1}^{(m)} \leftarrow \mathbf{x}_k, \forall w \in [W]$.
6:     Set $S_k = \lfloor \frac{\eta_k}{\theta_k} \rfloor$.
7:     **while** $s < S_k$ **do**
8:         **for** $w = 1$ **to** $W$ **do**    *//can be executed in parallel*
9:             Compute gradients $\nabla f(\bar{\mathbf{x}}_{s+w-1}^{(m)})$.
10:         **end for**
11:         **for** $w = 1$ **to** $W$ **do**    *//can be executed in parallel*
12:             $\bar{\mathbf{x}}_{s+w}^{(m+1)} = \bar{\mathbf{x}}_s^{(m)} - \theta_k^2 \sum_{j=s}^{s+w-1} (s + w - j) \nabla f(\bar{\mathbf{x}}_j^{(m)})$.
13:         **end for**
14:         **for** $w = 1$ **to** $W$ **do**    *//can be executed in parallel*
15:             Calculate relative error
                $e_{s+w} = \|\bar{\mathbf{x}}_{s+w}^{(m+1)} - \bar{\mathbf{x}}_{s+w}^{(m)}\| / \|\bar{\mathbf{x}}_{s+w}^{(m)}\|$.
16:         **end for**
17:         Update the stride $R$ of the window:
            $R \leftarrow \min(W, \min_{j \in [W]:e_{s+j} > \tau} j)$.
18:         **for** $r = 1$ **to** $R$ **do**
19:             Init points that the window will cover at the next
            inner iteration: $\bar{\mathbf{x}}_{s+W+r}^{(m+1)} \leftarrow \bar{\mathbf{x}}_{s+W}^{(m+1)}$.
20:         **end for**
21:         Update $s \leftarrow s + R$ and $m \leftarrow m + 1$.
22:         Update $W \leftarrow \min(W, S_k - s)$.
23:     Set $\mathbf{x}_{k+1} \leftarrow \bar{\mathbf{x}}_{S_k}^{(m+1)}$ to initialize the update at the next outer iteration $k + 1$.
24: **end for**
25: **Output:** $\mathbf{x}_{K+1}$.

---

The idea of parallelizing the steps in Picard iteration was recently applied to the task of quickly generating images by diffusion models (Shih et al., 2024), where obtaining an image from a diffusion model involves solving an ordinary differential equation. Motivated from their results, we consider parallelizing the steps in Hamiltonian Flow for optimization. However, when the number of steps $S$ is large, maintaining multiple $\bar{\mathbf{x}}_s^{(m)}$ at different steps $s$ becomes memory consuming. Hence, we adapt a sliding-window approach in Shih

et al. (2024), where only the steps of the flow in a current window are updated, and the window moves forward along the flow at each $m$. After the updates of the the steps in the current window, the relative error, i.e., $\|\bar{\mathbf{x}}_s^{(m+1)} - \bar{\mathbf{x}}_s^{(m)}\| / \|\bar{\mathbf{x}}_s^{(m)}\|$, for all steps $s$ in the window is calculated, which determines the stride of the window for the next iteration. Algorithm 5 details the proposed method.

# 4 Numerical results

We experiment with the proposed methods of frictionless Hamiltonian Descent via discretization and parallel optimization and report their performance. Specifically, Subsection 4.1.1 reports the empirical results of HD-LF (Algorithm 2 with Leapfrog) and HD-ST (Algorithm 2 with Stratified Integrator) under four different forms of kinetic energy $\phi(\cdot)$ as well as their counterparts such as GD, Normalized GD, Sign GD, and Coordinate Descent, with and without Heavy Ball (HB) momentum for multinomial logistic regression, while Subsection 4.2 provides results of neural network training. Appendix B.1 and B.2 include more experimental results of minimizing a non-convex function and matrix sensing.

In subsection 4.3, we evaluate the performance of the proposed HD via Parallel Picard Iteration (Algorithm 5). We primarily display results of multinomial logistic regression on different classification datasets. Our results clearly shows consistent speedup across the datasets. Appendix B.3 also displays the speedup across different cases of the kinetic energy (i.e., $\psi(\mathbf{v}) = \frac{1}{2}\|\mathbf{v}\|_2^2, \|\mathbf{v}\|_2, \|\mathbf{v}\|_1$, and $\|\mathbf{v}\|_\infty$).

Table 1: Detailed information of the datasets.

| DATASET | ALL SAMPLES | TRAINING SET | TEST SET | # OF FEATURES |
|---------|-------------|--------------|----------|---------------|
| A8A | 32561 | 22696 | 9865 | 123 |
| A9A | 48842 | 32561 | 16281 | 123 |
| MNIST | 60000 | 50000 | 10000 | 784 |
| USPS | 9298 | 7291 | 2007 | 256 |

The details of the datasets used for the experiments are described in Table 1[1]. The experiments are conducted on an Amazon EC2 P3 instance with 4 Tesla V100 GPUs. The code for reproducing the experiments is available on `https://github.com/oyhah/ParallelHD` and can also be accessed via the link in the supplementary material.

## 4.1 Hamiltonian Descent with a Geometric Numerical Integrator

### 4.1.1 MNIST dataset

Figure 2 and Figure 3 compare HD-LF, HD-ST, and their counterparts with different forms of the kinetic energy for the task of multinomial logistic regression. When the kinetic energy $\phi(\mathbf{v})$ is the squared $l_2$ norm, the counterparts of HDs are GD and HB (Polyak, 1964), where the former is without momentum and the latter is a momentum method. Similarly, when $\phi(\mathbf{v}) = \|\mathbf{v}\|_2$, the baselines of HDs are Normalized GD and Normalized GD with HB dynamics (Normalized GD-HB); when $\phi(\mathbf{v}) = \|\mathbf{v}\|_1$, the comparators are Sign GD and Sign GD with HB momentum (Sign GD-HB); and for the other form of $\phi(\mathbf{v}) = \|\mathbf{v}\|_\infty$, Coordinate Descent and Coordinate Descent with HB momentum (CD-HB) serve as the counterparts of HDs. For HD-LF and HF-ST with four forms of kinetic energy, we report their performance under the specification where the step size $\theta^2$ is set to $\theta^2 = 0.01$ and the number of integrator steps $S$ is $S = 10$ for all the iterations in Algorithm 2. For the counterparts of HDs, the step size is $\eta = 0.01$, which is the best from the set $[0.001, 0.01, 0.1]$, and the value of the momentum parameter is $\beta = 0.9$. The parameter $\theta^2$ is set to 0.01, since Lemma 2 shows that HD is equivalent to GD with a step size of order $O(\theta^2)$. Note that in all experiments we use the same initialization for compared algorithms but we plot the results after one round of iteration, not from the initialization.

---

[1]All the datasets are available at `https://www.csie.ntu.edu.tw/~cjlin/libsvmtools/datasets/`

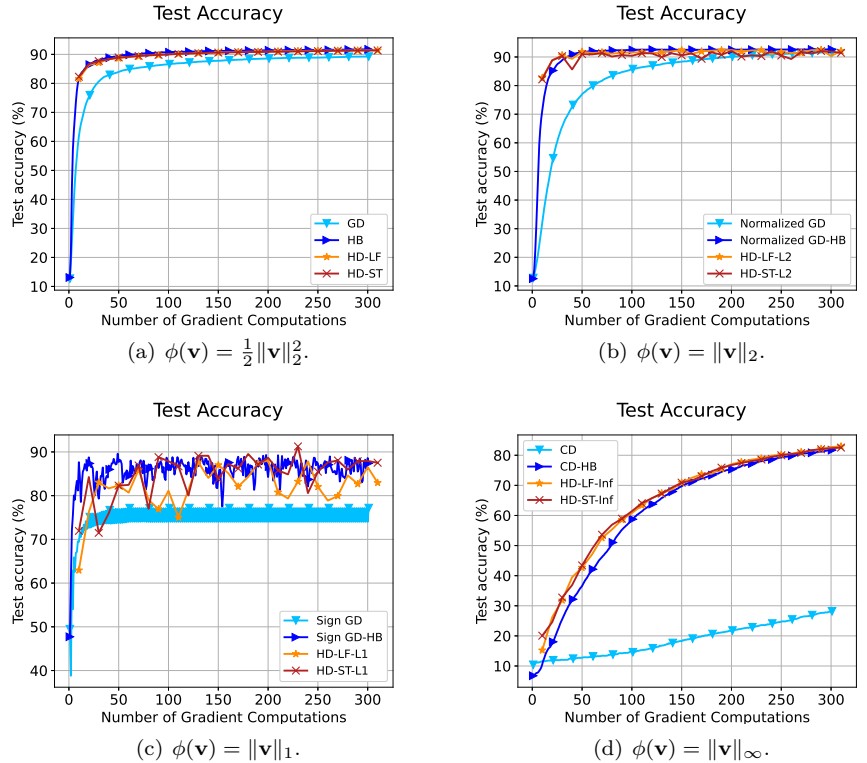

Figure 2: Test accuracy on MNIST dataset. Each line corresponds to HD-LF, HD-ST, or their counterparts with and without HB momentum, under four different forms of the kinetic energy $\phi(\cdot)$.

As HD has inner steps for simulating the Hamiltonian Flow at each iteration $k$, we report the testing accuracy and training loss vs. the total number of gradient computations in Figure 2 and Figure 3 to have a fair comparison. Specifically, we plot the progress of the algorithms up to 300 number of gradient computations; in the case of HDs, this corresponds to $K = 30$ iterations with $S = 10$ steps to simulate the flow at each iteration $k$. From both figures, HD-LF and HD-ST consistently outperform their counterparts without momentum under different cases of the kinetic energy except when $\phi(\mathbf{v}) = \|\mathbf{v}\|_\infty$ (i.e., Sign GD). Furthermore, both HD-LF and HD-ST have competitive performance with their counterparts that incorporate the momentum dynamics. We note that in the literature, Sign GD does not necessarily converge to a global optimal point in convex optimization, see e.g., Karimireddy et al. (2019) for an example of a provable failure of Sign GD. This is aligned with our observation in Subfigure (c) of both figures, where the algorithms seemingly converge to different points. Interestingly, HD-LF and HD-ST (and Sign GD with HB momentum) converge faster and achieve higher test accuracy than Sign GD, while having higher training loss compared to Sign GD.

## 4.2 Training ResNet on CIFAR10 Dataset

We further conducted experiments on training a ResNet18 model on CIFAR10 dataset with Hamiltonian Descent. We still use Euler discretization and kinetic function $\phi(\mathbf{v}) = \frac{1}{2}\|\mathbf{v}\|^2$ in HD and compare it with GD and HB.

In the experiments we use stochastic gradient rather than the full gradient. We actually implement a stochastic version of Hamiltonian Descent and compare it with SGD and SGD with Momentum. To simplify, we still use HD, GD and HB to represent these algorithms. The learning rate of these algorithms is 0.01 and the batch size is 64. The momentum of HB is 0.9 and the integration time is 0.2. Fig. 4(a) and (b) show the testing accuracy and training loss of the HD, GD and HB algorithms. The HD can get similar final performance as GD and HB, meaning it can be effectively applied to neural network training. Fig. 4(c) and

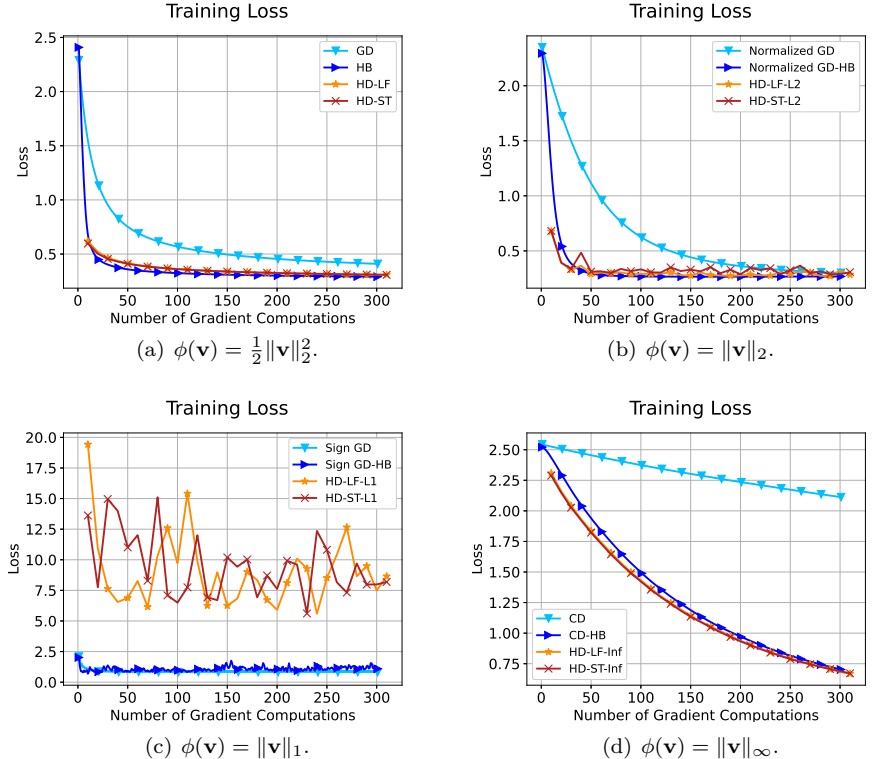

Figure 3: Training loss on MNIST dataset. Each line corresponds to HD-LF, HD-ST, or their counterparts with and without HB momentum, under four different forms of the kinetic energy $\phi(\cdot)$.

Table 2: Execution time of frictionless HD via Parallel Picard Iteration with 4 GPUs ($\phi(\mathbf{v}) = \frac{1}{2}\|\mathbf{v}\|_2^2$).

| RUNNING TIME | A8A | A9A | MNIST | USPS |
|---|---|---|---|---|
| 1 GPU | 93.2s | 99.6s | 403.1s | 83.1s |
| 4 GPUs | 47.7s | 50.3s | 128.9s | 42.7s |
| SPEEDUP | 1.95x | 1.98x | 3.13x | 1.95x |

(d) display the testing accuracy and training loss of HD with different integration time. It is seen for HD with stochastic gradient, the impact of integration time is not obvious. We can easily choose an integration time to get an acceptable performance.

## 4.3 Hamiltonian Descent via Parallel Picard Iteration

Figure 5 and 6 compare the proposed frictionless Hamiltonian Descent with Parallel Picard Iteration (Algorithm 5) when using 1 GPU, 2 GPUs, and 4 GPUs, where the kinetic energy $\phi(\mathbf{v}) = \frac{1}{2}\|\mathbf{v}\|_2^2$. The step size $\theta$ is 0.1, the number of integrator steps $S$ is 20, and the threshold for relative error $\tau$ is 0.001. The sliding window size corresponds to the number of GPUs used in our experiments. Choosing the threshold $\tau$ can be a delicate process: a larger error threshold reduces the accuracy of the Picard iteration approximation, as convergence is not fully achieved within each window, whereas a smaller threshold enforces more iterations within a single window, significantly increasing the computational cost. In our experiment, we choose the threshold that is the best among the set $[0.0001, 0.001, 0.01, 0.1]$.

Both figures show consistent speedup across all datasets when multiple GPUs are used to execute the proposed algorithm. A significant speedup is observed when using 2 GPUs compared to 1 GPU, and similarly, even greater speedup is observed with the use of 4 GPUs. In addition, we conduct the experiments for

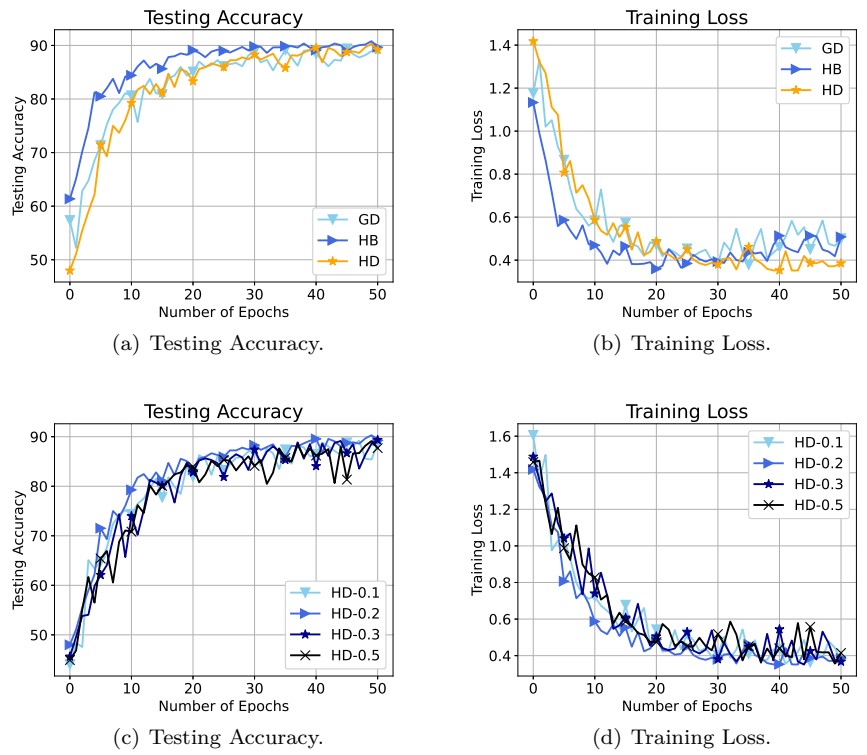

Figure 4: Testing Accuracy and Training Loss of neural network training with different algorithms and parameters.

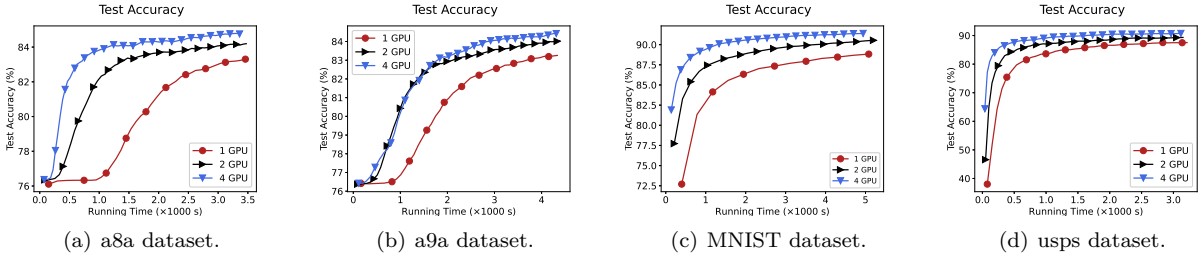

Figure 5: Testing accuracy vs. running time (seconds) on the four datasets of HD via Parallel Picard Iteration (Algorithm 5) with $\phi(\mathbf{v}) = \frac{1}{2}\|\mathbf{v}\|_2^2$.

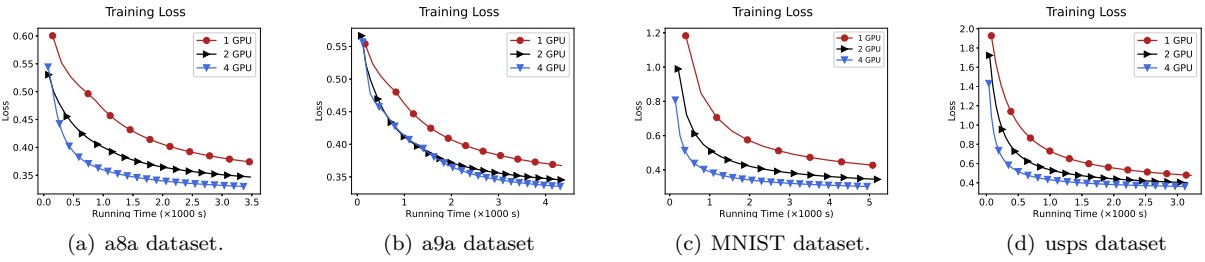

Figure 6: Training loss vs. running time (seconds) on the four datasets of HD via Parallel Picard Iteration=(Algorithm 5) with $\phi(\mathbf{v}) = \frac{1}{2}\|\mathbf{v}\|_2^2$.

different forms of the kinetic energy and report the results in the supplementary, where we also observe that with the use of multiple GPUs, Hamiltonian Descent via Parallel Picard Iteration reduces the actual running time, compared to the case of a single GPU. We note that due to the criteria for updating the stride of the sliding window in each inner iteration of Algorithm 5, the actual number of steps when approximating the Hamiltonian flow in the inner loop of Algorithm 5 can vary across different datasets, resulting in varying speedup via multiple GPUs across different datasets. Though the speedup can vary, we found that the proposed HD via Parallel Picard Iteration is effective in accelerating the process of simulating a flow in the experiments.

## 5 Conclusions

In this paper, we make the idea of frictionless HD more applicable by presenting different forms of kinetic energy and considering frictionless HD with the Leapfrog integrator and the Stratified Monte Carlo integrator. We found some interesting connections of the resulting dynamics to the existing ones and also some novel update schemes. We further propose speeding up approximately simulating the flow via parallel optimization and show exciting empirical results of the acceleration via multiple GPUs. Nevertheless, a couple of directions remain open in this work. First, Frictionless HD with Parallel Picard Iteration (Algorithm 5) requires $W$ copies of the model to be loaded into memory in order to perform parallel-in-time updates, where $W$ is the window-size parameter. This results in a significant memory footprint, especially for large models, which in turn may limit the potential applications of the proposed algorithm. How to elegantly reduce this seemingly inherent and substantial memory cost is therefore a worthwhile direction for future research. Any concrete and effective approach to reducing the memory footprint could substantially improve the applicability of the algorithm to large-scale model training, such as large language models, which typically rely on large transformer architectures. Second, while we show that discretizing the Hamiltonian flow with multiple integration steps induces different momentum-like dynamics under different choices of kinetic energy, potentially highlighting the modularity of the frictionless Hamiltonian Descent framework. When and how these new optimization dynamics yield tangible performance benefits remain underexplored, especially in the context of non-convex optimization.

## Acknowledgements

The authors thank the anonymous reviewers and the action editor for their useful comments. JW also appreciates Andre Wibisono for the discussions. This work is supported by NSF CCF-2403392.

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

## A Proof of Lemma 2

*Proof.* When $\phi(\mathbf{v}) = \|\mathbf{v}\|_2$, we have $\nabla_{\mathbf{x}} H(\mathbf{x}, \mathbf{v}) = \nabla f(\mathbf{x})$ and $\nabla_{\mathbf{v}} H(\mathbf{x}, \mathbf{v}) = \frac{\mathbf{v}}{\|\mathbf{v}\|_2}$. Similarly, From Algorithm 3, with the initial condition $\bar{\mathbf{x}}_0 \leftarrow \mathbf{x}_k$ and $\bar{\mathbf{v}}_0 \leftarrow 0$, one can see that the dynamic with a single Leapfrog step is:

$$\bar{\mathbf{v}}_{\frac{1}{2}} = \bar{\mathbf{v}}_0 - \frac{\theta}{2}\nabla f(\bar{\mathbf{x}}_0) = -\frac{\theta}{2}\nabla f(\mathbf{x}_k)$$

$$\bar{\mathbf{x}}_1 = \bar{\mathbf{x}}_0 + \theta \frac{\bar{\mathbf{v}}_{\frac{1}{2}}}{\|\bar{\mathbf{v}}_{\frac{1}{2}}\|} = \mathbf{x}_k - \theta \frac{\nabla f(\mathbf{x}_k)}{\|\nabla f(\mathbf{x}_k)\|}. \tag{15}$$

With only a single step, Leapfrog outputs $\bar{\mathbf{x}}_1$, which is the position of the update $\mathbf{x}_{k+1}$ at the next iteration $k+1$. We hence obtain the update of Normalized GD, i.e., $\mathbf{x}_{k+1} = \mathbf{x}_k - \theta \frac{\nabla f(\bar{\mathbf{x}}_k)}{\|\nabla f(\bar{\mathbf{x}}_k)\|}$.

When $\phi(\mathbf{v}) = \|\mathbf{v}\|_1$, we have $\nabla_{\mathbf{x}} H(\mathbf{x}, \mathbf{v}) = \nabla f(\mathbf{x})$ and one subgradient is $\partial_{\mathbf{v}} H(\mathbf{x}, \mathbf{v}) = \text{sign}(\mathbf{v})$. Similarly, we can see that the dynamic with a single Leapfrog step is:

$$\bar{\mathbf{v}}_{\frac{1}{2}} = \bar{\mathbf{v}}_0 - \frac{\theta}{2} \nabla f(\bar{\mathbf{x}}_0) = -\frac{\theta}{2} \nabla f(\mathbf{x}_k)$$
$$\bar{\mathbf{x}}_1 = \bar{\mathbf{x}}_0 + \theta \, \text{sign}(\bar{\mathbf{v}}_{\frac{1}{2}}) = \mathbf{x}_k - \theta \text{sign}(\nabla f(\mathbf{x}_k)). \tag{16}$$

With only a single step, Leapfrog outputs $\bar{\mathbf{x}}_1$, which is the position of the update $\mathbf{x}_{k+1}$ at the next iteration $k+1$. We hence obtain the update of Sign GD, i.e., $\mathbf{x}_{k+1} = \mathbf{x}_k - \theta \text{sign}(\nabla f(\mathbf{x}_k))$.

When $\phi(\mathbf{v}) = \|\mathbf{v}\|_\infty$, we have $\nabla_{\mathbf{x}} H(\mathbf{x}, \mathbf{v}) = \nabla f(\mathbf{x})$ and one subgradient is $\partial_{\mathbf{v}} H(\mathbf{x}, \mathbf{v}) = \text{sign}\left(\mathbf{v}[i_{\max}]\right) \mathbf{e}_{i_{\max}}$, where $i_{\max} \in \arg\max_{i \in [d]} |\mathbf{v}[i]|$, since $\|\mathbf{v}\|_\infty$ is the absolute value of the element in $\mathbf{v}$ with largest absolute value. And $\mathbf{e}_{i_{\max}}$ is the one-hot vector with index $i_{\max}$ being non-zero. Similarly, we can see that the dynamic with a single Leapfrog step is:

$$\bar{\mathbf{v}}_{\frac{1}{2}} = \bar{\mathbf{v}}_0 - \frac{\theta}{2} \nabla f(\bar{\mathbf{x}}_0) = -\frac{\theta}{2} \nabla f(\mathbf{x}_k)$$
$$\bar{\mathbf{x}}_1 = \bar{\mathbf{x}}_0 + \theta \, \text{sign}(\bar{\mathbf{v}}_{\frac{1}{2}}) = \mathbf{x}_k - \theta \, \text{sign}\left(\nabla f(\mathbf{x}_k)[i_{\max}]\right) \mathbf{e}_{i_{\max}}. \tag{17}$$

With only a single step, Leapfrog outputs $\bar{\mathbf{x}}_1$, which is the position of the update $\mathbf{x}_{k+1}$ at the next iteration $k+1$. We hence obtain the update of Coordinate Descent, i.e., $\mathbf{x}_{k+1} = \mathbf{x}_k - \theta \, \text{sign}\left(\nabla f(\mathbf{x}_k)[i_{\max}]\right) \mathbf{e}_{i_{\max}}$, where $i_{\max} \in \arg\max_{i \in [d]} |\nabla f(\mathbf{x}_k)[i]|$. □

## B  Additional Experiments

### B.1  Ackley function

We also conduct an experiment for minimizing the Ackley function, which is known to be a highly non-convex function with two variables:

$$F(\mathbf{x}) = -20 \exp\left[-0.2\sqrt{0.5(x_1^2 + x_2^2)}\right]$$
$$- \exp\left[0.5(\cos 2\pi x_1 + \cos 2\pi x_2)\right] + e + 20. \tag{18}$$

In this experiment, we consider HD with the kinetic energy being $\phi(\mathbf{v}) = \frac{1}{2}\|\mathbf{v}\|_2^2$.

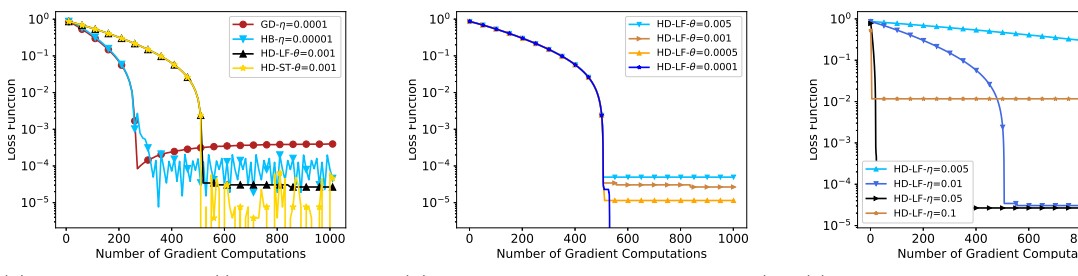

(a) Function value $F(\cdot)$ vs. # gradient computations.
(b) HDs with different step sizes $\theta$ (integration time $\eta$ fixed).
(c) HDs with different values of integration time $\eta$ (step size $\theta$ fixed).

Figure 7: Minimizing the Ackley function.

Figure 7.1(a) compare the proposed HDs with the baselines, GD and HB, starting from the same point. For HD-LF and HD-ST, the integration time is fixed as $\eta = 0.01$ and step size is chosen as $\theta = 0.001$. For GD

and HB, the step size $\eta$ is tuned on the grid $[10^{-1}, 10^{-2}, \ldots, 10^{-6}]$ for the best one. From the figure, we can see that HB outperforms GD, and our proposed HD-LF and HD-ST can further outperforms HB. We also observe that HD-LF behaves more stably than HB during the optimization process, while the update of HD-ST is less stable due to the randomness of the step size $\zeta_s$.

We then investigate the impact of parameter choices in HD-LF. Figure 7.1(b) shows the performance of HD-LF with different step size $\theta$ with a fixed integration time. The integration time $\eta$ is fixed as $\eta = 0.01$. Then, the number of integrator steps $S$ varies with $\theta$. With a smaller step size, the discretized flow tracks the exact Hamiltonian flow better and can converge to a lower value. However, the computational burden increases with more integrator steps within one execution of the Hamiltonian flow, suggesting the need to accelerate the process of simulating a Hamiltonian flow (and we propose HD via Parallel Optimization in Section 3 to address this issue). Figure 7.1(c) displays the performance of HD-LF with different value of the integration time $\eta$, while the step size is fixed as $\theta = 0.001$. When $\eta = 0.01$ or $0.05$, the performance of HD-LF is better than HD-LF with a smaller value than $0.01$ or a larger value than $0.05$, showing that there is a range of $\eta$ to obtain the best performance of HD-LF.

## B.2 Matrix Sensing

Low rank matrix sensing problem is a typical non-convex problem widely applied to image processing, multi-task regression, and metric embeddings (Davis et al., 2025). It aims to recover a symmetric positive semidifinite matrix $X \in \mathbb{R}^{d \times d}$ with low rank $r \ll d$ from a set of linear measurements $y_i = \langle A_i, X \rangle$, where $A_i \in \mathbb{R}^{d \times d}$ are known measurement matrices. Using low rank factorization of the matrix $X = BB^T$, where $B \in \mathbb{R}^{d \times k}$, we can formulate the optimization problem as:

$$\min_{B \in \mathbb{R}^{d \times k}} \frac{1}{4m} \sum_{i=1}^{m} \left( y_i - \langle A_i, BB^T \rangle \right)^2, \tag{19}$$

where $m$ is the number of measurements. The rank $k$ of matrix $B$ is usually larger than $r$ since we do not know the exact value of $r$ in practice and we choose an overestimated value $k$. In the experiments $A_i$ is generated from $A_i = a_i a_i^T - \tilde{a}_i \tilde{a}_i^T$, where $a_i$ and $\tilde{a}_i^T$ are standard Gaussian vectors.

In this matrix sensing problem, we test our proposed Hamiltonian Descent (HD) algorithm and compare it with Gradient Descent (GD) and Heavy Ball (HB) methods. To simplify, we use Euler discretization and kinetic function $\phi(\mathbf{v}) = \frac{1}{2}\|\mathbf{v}\|^2$ in HD.

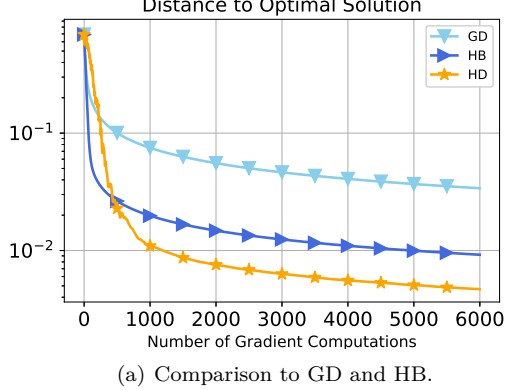
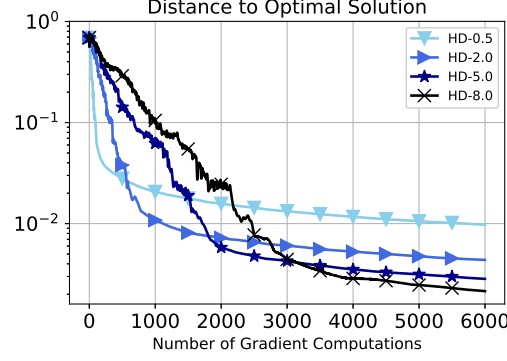

(a) Comparison to GD and HB.

(b) HDs with different values of integration time $\eta$.

Figure 8: Distance to the optimal solution in matrix sensing experiments.

In the experiments, the exact rank $r$ is 2 and the overparameterized rank $k$ is 4. The dimension of matrix $d$ is 100, and number of measurements $m$ is 1000. Fig. 8 (a) shows the performance of HD and the comparison to GD and HB. The learning rate of these algorithms are 0.05. For HD, the integration time is 2, with 40 inner steps per iteration. For HB, the momentum is 0.9. We can see the HD can outperform GD and HB

with lower distance to the optimal solution. Fig. 8 (b) shows HD performance with different integration time values $\eta$ as 0.5, 2.0, 5.0, 8.0. When the integration time is larger, the final distance to the optimal solution can be smaller, while the convergence is slower.

### B.3 Parallel Optimization with Different Kinetic Energy

We also provide experimental results of Algorithm 5 with different forms of the kinetic energy in Figure 9, where we also observe that with the use of multiple GPUs, Hamiltonian Descent via Parallel Picard Iteration reduces the actual running time, compared to the case of a single GPU.

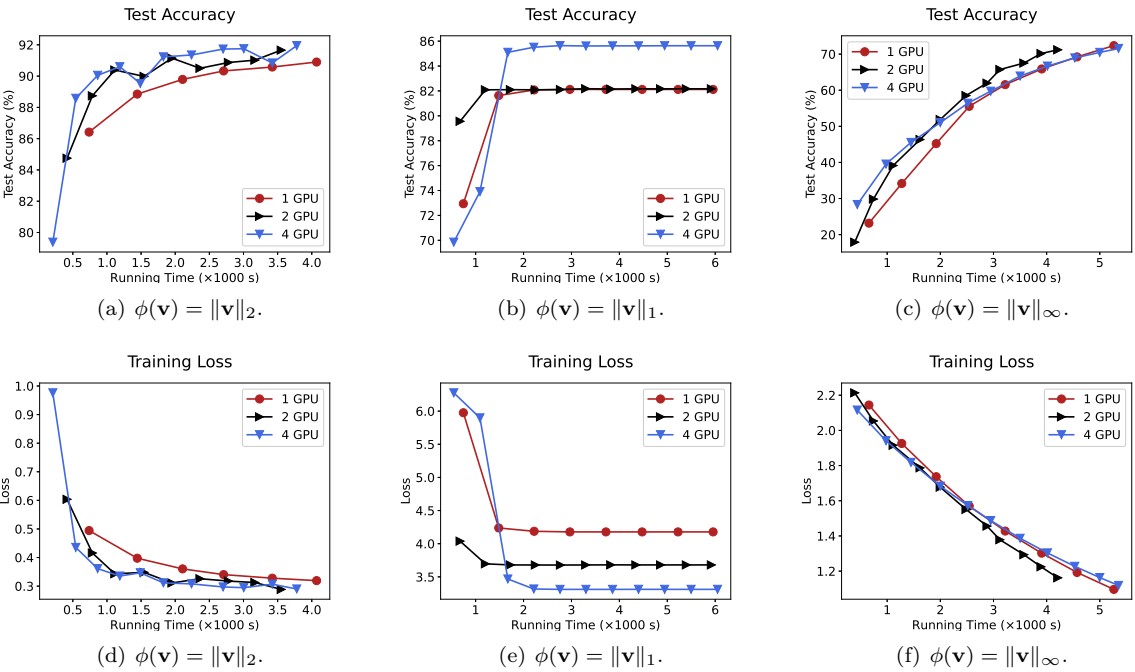

Figure 9: Testing Accuracy vs. running time (seconds) on MNIST of HD via Parallel Picard Iteration (Algorithm 5) under the cases of different forms of kinetic energy $\phi(\cdot)$.

