# OpenReview forum: "Frictionless Hamiltonian Descent with Discretization and Parallel Optimization"
_TMLR — Accepted by TMLR_

### Review · Reviewer_oHeX · 2025-12-21

**Summary Of Contributions:**

The paper proposes an extension to Frictionless Hamiltonian Descent (HD) by introducing geometric numerical integrators—specifically the Leapfrog integrator and the Stratified Monte Carlo integrator—to simulate Hamiltonian flow, replacing the requirement for an exact flow. The authors generalize the framework by introducing various forms of kinetic energy (e.g., squared $\ell_2$, $\ell_1$, and $\ell_\infty$ norms).

A key theoretical contribution is presented in Lemma 2, demonstrating that executing a single step of the Leapfrog integrator with these specific kinetic energies recovers standard optimization algorithms such as Gradient Descent, Normalized GD, Sign GD, and Coordinate Descent. Furthermore, to address the computational bottleneck of sequential updates in Hamiltonian flow, the authors propose a parallelization technique using Picard Iteration. This allows for simultaneous gradient computations across multiple GPUs. Experiments are conducted on multinomial logistic regression, neural network training (ResNet18 on CIFAR10), and non-convex problems (Ackley function, Matrix Sensing), demonstrating that the proposed method is competitive and that the parallelization yields significant speedups (approx. 2-3x).

**Additional Comments:**

N/A

**Audience:**

Yes

**Audience Explanation:**

There are at least two distinct sub-communities within machine learning that would find this paper of interest: The theoretical contributions of this paper are likely to interest those studying the foundations of optimization algorithms. On the other hand, by proposing a "Parallel Picard Iteration" method that leverages multiple GPUs to parallelize these steps, the paper offers a concrete engineering solution that achieves measured speedups (2-3x) on standard hardware. This aspect is relevant to researchers focused on hardware efficiency and distributed training.

**Claims And Evidence:**

Yes

**Claims Explanation:**

1. Theoretical Evidence: The authors' claim that their framework unifies various optimization algorithms is rigorously supported by Lemma 2.
2. Empirical Evidence for Optimization Performance: The claim that the proposed Frictionless Hamiltonian Descent (HD) is a competitive optimization method is supported by a diverse set of experiments.
3. Evidence for Parallelization Speedup: The claim that the proposed Parallel Picard Iteration (Algorithm 5) accelerates training is backed by clear, quantitative data. Table 2 and Figures 5-6 demonstrate consistent wall-clock time reductions across multiple datasets.

**Requested Changes:**

1. In Section 4.1 (Ackley function), the paper explicitly states that the step size for baselines (GD and HB) was tuned on a grid. However, the tuning protocol for the Multinomial Logistic Regression (Section 4.1.1) and Neural Network (Section 4.2) experiments is less explicit regarding the baselines. To ensure the comparison is rigorous, please add a brief description (either in the main text or Appendix) detailing the tuning range and selection criteria used for the baselines in these sections.
2. Section 2 provides the update equations for Hamiltonian Descent when using multiple Leapfrog steps ($S > 1$) 2. These equations introduce "novel momentum-like dynamics" involving weighted accumulations of gradients. To improve readability and theoretical depth, please expand the discussion in this section to provide more intuition on why these specific dynamics are beneficial. For instance, how does the weighting of gradients in Equation 6 differ qualitatively from the accumulation in standard Heavy Ball momentum?
3. Algorithm 5 introduces a "sliding window" approach with window size $W$ and error threshold $\tau$. Please add a brief clarification (likely in Section 3 or Appendix) regarding how these parameters were selected for the experiments and how sensitive the algorithm's performance or stability is to choices of $W$ and $\tau$.
4. Please proofread the manuscript for minor typos. For example:
- "runnning time": the 2nd line from the bottom of section1
- "singe GPU": the 5th line from the bottom of section 4; the 3rd line of section B.3

---

> ### Author Response · Authors · 2026-01-06
>
> Thanks for your positive comments and constructive suggestions. Please find our point-to-point responses below.
>
> ### 1. [Step Size Tuning in Logistic Regression.]
>
> Thanks for your suggestion. For logistic regression, we tune the step size of Gradient Descent and other baselines in $[0.001, 0.01, 0.1]$ and $0.01$ is the best. As stated in Lemma 2, the HD is equivalent to GD with step size as $O(\theta^2)$, thus we also choose $\theta^2 =0.01$ in the experiments for fair comparison. We have explicitly stated our choice of step size in the revised version. We performed a more precise search of step size on Ackley function minimization since it can show the difference between HD and conventional GD and HB algorithms more clearly, compared to logistic regression and neural network training.
>
> ### 2. [Dynamics for $S>1$.]
>
> Thanks for your suggestion. In our preliminary experiments, we observe a glimpse of the potential benefits of Frictionless Hamiltonian Descent for non-convex optimization, as illustrated by experiments on the Ackley function. In particular, Hamiltonian Descent reaches points with lower objective values that are not attained by Heavy-Ball methods, even when the step size of the latter is carefully tuned. However, we acknowledge that the theoretical foundations of Frictionless Hamiltonian Descent in non-convex optimization remain to be developed, and we leave this as future work.
>
> ### 3. [Choice of Sliding Window and Error Threshold.]
>
> In our experiments of Parallel-in-Time Frictionless Hamiltonian Descent, the sliding window size is limited by the number of GPUs.
> The window size was set to 4 simply because we have 4 GPUs. It is possible to achieve more speed-up if one has more computational resources (with a larger window size accordingly).
>
> For the error threshold, a larger error threshold might reduce the accuracy of approximation of Picard iteration, and a smaller threshold might enforce more iterations within one window. In other words, there is potentially a trade-off. In experiments we tried threshold as 0.0001, 0.001, 0.01, 0.1 and chose the best. We have added this discussion at Section 4.3 in the revised version.
>
> ### 4. [Typos.]
>
> Thanks for your careful reading. We have corrected the typos.

---

> > ### Comment · Reviewer_oHeX · 2026-01-12
> >
> > I've carefully read the authors' responses to my feedback and have no further questions.

---

### Review · Reviewer_nmHB · 2025-12-28

**Summary Of Contributions:**

The paper addresses the challenge of applying Hamiltonian Descent to general non-quadratic optimization problems, where the Hamiltonian flow lacks a closed-form solution and existing approaches such as Idealized Frictionless Hamiltonian Descent are therefore inapplicable. To overcome this limitation, the authors propose a discretized formulation of the Hamiltonian flow and employ state-of-the-art numerical integrators to approximate the resulting dynamics.

The first contribution is a systematic study of alternative kinetic energy formulations, including squared- $\ell_2, \ell_2, \ell_1$, and $\ell_\infty $ norms. The authors show that when a single integrator step is used per iteration, the resulting updates resemble those of existing descent-based optimization methods, whereas using multiple integrator steps induces dynamics similar to momentum-based algorithms.

The second contribution is the use of Picard iteration to simulate the Hamiltonian flow, which, after discretization, enables parallel execution of the flow updates and allows efficient implementation on multiple GPUs.

The proposed approach is evaluated on certain non-convex functions and machine learning datasets. Experimental results report training loss and test accuracy comparisons against standard optimization methods. The results indicate that the proposed Frictionless Hamiltonian Descent with parallelization achieves performance competitive with momentum-based methods and outperforms non-momentum baselines. In addition, the paper evaluates the quantitative speedup obtained by increasing the number of GPUs during training and testing.

**Additional Comments:**

Section 4.3- In addition, we conduct the experiments $ \textbf{for} $ different...

**Audience:**

Yes

**Audience Explanation:**

The methodology and findings presented in the paper are of interest and may be relevant to researchers exploring alternative optimization algorithms for training neural networks.

**Broader Impact Concerns:**

This work does not appear to raise any significant broader impact or ethical concerns.

**Claims And Evidence:**

Yes

**Claims Explanation:**

The update schemes corresponding to the different choices of kinetic energy are derived through theoretical analysis that is clear and easy to follow. Experimental results are presented across multiple settings, including non-convex benchmark functions and machine learning datasets. These results show improved performance over non-momentum baseline algorithms and competitive performance relative to momentum-based methods when training neural networks. Additionally, the proposed algorithm outperforms the heavy-ball method in the Ackley function and matrix sensing examples.

The use of parallel Picard iteration within the proposed framework is also clearly demonstrated, although the performance benefits of parallelization are not consistently evident across all datasets. Overall, the experimental results are largely representative of the claims made in the paper.

There are a few points where additional clarification would strengthen the presentation. First, providing details of the neural network architecture used for the MNIST experiments would improve reproducibility. Second, stating the final form of the objective function used in the matrix sensing example (Eq. 9) would be helpful.

**Requested Changes:**

While the methodology is technically sound, the paper would benefit from a clearer motivation for the practical usefulness of the proposed approach relative to well-established momentum-based optimization methods. In particular, the manuscript does not sufficiently articulate the scenarios in which the proposed method is expected to offer advantages. Including a dedicated discussion that clarifies the intended use cases, potential benefits, and scenarios where the method may outperform existing momentum-based algorithms would help place the contributions in a broader optimization context.

---

> ### Author Response · Authors · 2026-01-06
>
> Thanks for your positive comments and constructive suggestions. Please kindly find our point-to-point responses below.
>
> ### 1. [Neural Network Architecture.]
>
> On MNIST dataset, we have done the experiments with multinomial logistic regression, not neural network. To further illustrate the effectiveness of our methods, we also train ResNet18 model on CIFAR10 dataset.
>
> ### 2. [Objective Function in Matrix Sensing.]
>
> As stated in the manuscript, the objective function in low-rank matrix sensing is
> \begin{align}
>     \min\limits_{B \in \mathbb{R}^{d \times k}} \frac{1}{4m} \sum_{i=1}^m \left( y_i - \left< A_i, B B^\top \right> \right)^2,
> \end{align}
> where $m$ is the number of measurements, $B$ is a low-rank matrix with rank $k$. The variable to be optimized is exactly the matrix $B$. $A_i$ is generated from $A_i = a_i a_i^\top - \tilde{a}_i \tilde{a}_i^\top$, where $a_i$ and $\tilde{a}_i^\top$ are standard Gaussian vectors, and $y_i$ is generated from $y_i = \left< A_i, X \right> = \left< A_i, B B^\top \right> $. We have added a more clear expression on the revised manuscript.
>
> ### 3. [Comparison to Momentum Method.]
>
> We found that discretizing the Hamiltonian flow with multiple integration steps leads to different momentum-like dynamics under different choices of kinetic energy, which may shed light on the potential modularity of the frictionless Hamiltonian Descent framework. Nevertheless, the reviewer is correct that when and how these new optimization dynamics provide tangible benefits remain open questions; our preliminary experimental results showed that they are competitive with Heavy Ball. We have updated the conclusion to acknowledge this limitation and view it as a possible direction for future work. On the other hand, we also wanted to clarify that the motivation of our work is to advance Frictionless Hamiltonian Descent. To this end, our work explores techniques for simulating Hamiltonian Flows and provides a parallel-in-time approach to simulate Hamiltonian Flow that leverages multiple GPUs to accelerate the optimization process.

---

> > ### Comment · Reviewer_nmHB · 2026-01-14
> >
> > The authors have adequately addressed all my concerns. I have no further questions at this time.

---

### Review · Reviewer_9Mhf · 2025-12-29

**Summary Of Contributions:**

The paper extends the Frictionless Hamiltonian Descent algorithm, originally proposed in Wang (2024), which leverages Hamiltonian flows to minimize unconstrained optimization problems $\min_{x \in \mathbb{R}^d} f(x)$. Unlike traditional momentum-based methods that incorporate friction to dissipate energy, HD relies on energy conservation with periodic resets of kinetic energy, ensuring descent without damping. Key advancements include:

- **Discretization via Geometric Numerical Integrators:** The idealized HD requires exact simulation of the Hamiltonian flow, which is infeasible for non-quadratic functions. The authors address this by employing two geometric integrators to approximate the flow:
The Leapfrog integrator, a symplectic method that maintains bounded energy errors.
The Stratified Monte Carlo integrator, which introduces randomness and is shown to outperform Leapfrog in related sampling contexts.
These lead to novel discrete-time optimization dynamics. For instance, with a single integration step, the resulting algorithms recover established methods such as Gradient Descent, Normalized Gradient Descent, Sign Gradient Descent, and Coordinate Descent, depending on the form of the kinetic energy $\phi(v)$ in the Hamiltonian $H(x, v) = f(x) + \phi(v)$. With multiple steps, new momentum-like updates emerge, accumulating gradients along the flow without friction-based decay.

- **Generalization of the Hamiltonian:** The authors broaden the framework by considering various non-negative kinetic energy functions $\phi(v)$, where $\phi(v) = 0$ only when $v = 0$. This yields a family of algorithms, providing a unified perspective on optimization design and revealing connections to existing methods.

- **Parallel Optimization Technique:** To accelerate the sequential nature of flow simulation, the paper introduces a parallelization method based on Picard Iteration, a fixed-point technique for solving differential equations. This enables simultaneous gradient computations across multiple GPUs via iterative updates, approximating the discretized flow. Theoretically, after sufficient iterations, it exactly matches the sequential output. Empirically, it achieves 2-3x speedups in runtime for multinomial logistic regression on datasets like a8a and MNIST when using 4 GPUs compared to a single GPU.

**Additional Comments:**

This paper provides a valuable contribution to the literature and has theoretical novelty. The numerical results are good. Implementation details are provided. Hence I recommend accept if authors can address my concerns.

**Audience:**

Yes

**Audience Explanation:**

The paper introduces a novel extension of Hamiltonian-based optimization, addressing practical challenges in simulating flows for non-quadratic functions through discretization and parallelization, which holds clear relevance for the TMLR community focused on optimization theory and algorithms.

**Claims And Evidence:**

Yes

**Claims Explanation:**

Yes, the claims are well-supported by a combination of theoretical derivations and empirical evaluations. The theoretical contributions, such as the generalization of the Hamiltonian framework and the descent properties derived from energy conservation, are substantiated through clear lemmas and mathematical proofs that demonstrate the preservation of key properties like non-negative kinetic energy and equivalence to established algorithms under specific conditions. The discretization using geometric integrators (Leapfrog and Stratified Monte Carlo) is rigorously analyzed, with derivations showing how single-step approximations recover methods like Gradient Descent and multi-step versions yield novel momentum dynamics. Empirically, the parallel optimization technique via Picard Iteration is validated through experiments on multinomial logistic regression across datasets such as a8a and MNIST.

**Requested Changes:**

My primary concern is the absence of modern baselines and architectures in the experimental evaluation. The comparisons are primarily against standard gradient descent and heavy ball methods. In the context of deep learning and non-convex optimization, the standard baseline is Adam or AdamW (although they are variants of SignGD). Omitting adaptive gradient methods makes it difficult to assess the algorithm's competitiveness in modern ML workflows. Another point is that optimization papers in 2024/2025 are often expected to demonstrate effectiveness on Transformer architectures (e.g., GPT variants) to demonstrate they aren't limited to toy problems or older architectures like ResNet. Since LLM training is more memory-bounded, maintaining multiple steps for Picard Iteration is memory consuming and proposes a sliding window. An LLM experiment would be the perfect stress test to see if this sliding window approach is actually viable when the model states themselves are gigabytes in size.

---

> ### Author Response · Authors · 2026-01-06
>
> Thank you for your positive comments and helpful suggestions. On the one hand, we fully acknowledge that recent advances in machine learning are largely driven by large language models. On the other hand, the scope of our work is deliberately focused on advancing and enhancing Frictionless Hamiltonian Descent, in particular by exploring techniques for simulating Hamiltonian flows and by proposing a parallel-in-time approach to simulate Hamiltonian Flow that leverages multiple GPUs to accelerate the optimization process of Frictionless Hamiltonian Descent. Accordingly, the goal of this work was not to propose a method that competes with AdamW, Muon, or other state-of-the-art optimizers for training transformers or large language models.
>
> Nevertheless, we have updated the conclusion to acknowledge this limitation, and we have also made the weaknesses of the Parallel Picard Iteration more explicit --- it requires $W$ (the window-size parameter) copies of the model to be loaded into memory. For this reason, we are unable to perform large-scale LLM model training with the computational resources currently available to us. But we included training a ResNet on CIFAR 10 dataset and found supportive empirical evidence for the speedup.
> How to elegantly reduce this seemingly inherent and substantial memory cost is left as an open problem in this work. We also wish to emphasize that when the data exhibit a largely linear relationship between labels and features (e.g., house price prediction), linear models such as logistic regression, which we considered in our experiments, remain practical and widely used in machine learning and data science.

---

> > ### Comment · Reviewer_9Mhf · 2026-01-13
> >
> > Thank you for the detailed response. I don’t have any further questions.

---

### Comment · Action_Editor_fmF4 · 2026-01-01
**Discussion period**

Dear authors and reviewers,

We've entered the "discussion period", where the reviewers are now visible, and the authors can respond with a rebuttal (but it can be more than a single static rebuttal -- ideally we have a **dialogue**).  We have a bit under 2 weeks remaining for this phase.

Authors, you can also upload revisions to the paper.

Reviewers, when you view the rebuttal to your comments, please also take the time to read what the other reviewers said.  You can also bring things up privately (just don't check the "Everyone" or "TMLR Paper6496 Authors" boxes when you post a comment) if you want to discuss.

-Stephen (action editor for this paper)

---

### Decision · Action_Editor_fmF4 · 2026-02-13

**Recommendation:** Accept with minor revision

**Additional Comments:**

The reviewers are happy with the rebuttal and high-level changes, so I'm pleased to recommend acceptance.  Please make the following minor changes for the camera-ready version:

Below Eq (1), it's writen:
> (Wang, 2024) observes ...

which should be
> Wang (2024) observes ....

The issue is that you need to use the `\citet` and `\citep` commands appropriately. Use `\citep` when the citation is not a part of the grammar of the sentence, and use `\citet` when it is. Please correct this throughout the paper.

Also, make sure to capitalize proper nouns in the references. e.g., "Jun-Kun Wang. Hamiltonian descent and coordinate hamiltonian descent." should capitalize the second "h"; and "Ashia C. Wilson, Michael Jordan, and Benjamin Recht. A lyapunov analysis of momentum methods in optimization. JMLR, 2021" should capitalize the L in Lyapunov; etc.

Finally, thanks for using the blue text to mark your changes. Don't forget to revert that back to black text for the camera-ready version.

**Audience:**

Yes

**Audience Explanation:**

Yes. The topic of optimization, and Hamiltonan descent in particular, is generally of interest to readers. The results are significant enough that they are of interest.  The parallel Picard iteration is also itself of interest to some readers, especially as multiple GPU architectures are increasingly common.

**Claims And Evidence:**

Yes

**Claims Explanation:**

Yes. There is theory supporting the unification claim (Lemma 2), and empirical evidence via numerical simulations that the method is competetive on a range of problems, and that the parallelization can improve speed.  All reviewers agreed on this point.

---

> ### Author Response · Authors · 2026-02-23
> **Camera-ready version**
>
> We thank the AE for overseeing the review process of our manuscript and for soliciting constructive reviews. We have made minor revisions based on the AE’s helpful feedback:
>
> -We have updated the manuscript to ensure that \citep and \citet are used correctly in the camera-ready version.
>
> -We have corrected the references so that proper nouns are properly capitalized.
>
> Thank you!
>
> Best,
> Authors